# Cytokines, chemokines and growth factors profile in human aqueous humor in idiopathic uveitis

Marie-Hélène Errera[1,2,3]*, Ana Pratas[1], Sylvain Fisson[4], Thomas Manicom[1,2], Marouane Boubaya[5], Neila Sedira[1], Emmanuel Héron[1], Lilia Merabet[1], Alfred Kobal[1], Vincent Levy[5], Jean-Michel Warnet[6], Christine Chaumeil[1], Françoise Brignole-Baudouin[1,6], José-Alain Sahel[1,2,3], Pablo Goldschmidt[1], Bahram Bodaghi[2,7], Coralie Bloch-Queyrat[5]

1 Departments of Ophthalmology and Internal Medicine at Quinze-Vingts National Eye Hospital and DHU Sight Restore, Laboratory, Paris, France, 2 Sorbonne Universités, UPMC Univ Paris, Paris, France, 3 Department of Ophthalmology, UPMC Eye Center, University of Pittsburgh School of Medicine, Pennsylvania, United States of America, 4 Généthon, Inserm UMR_S951, Univ Evry, Université Paris-Saclay, EPHE, Evry, France, 5 Université Paris 13, Sorbonne Paris cité, INSERM U1163/CNRS ERL 8254, AP-HP, Hôpital Avicenne, URC-CRC GHPSS, Bobigny, France, 6 Faculty Pharmacy, Sorbonne Universities, Paris, France, 7 Pitié-Salpêtrière Hospital, DHU Sight Restore, Paris, France

* erreram@upmc.edu

**Data Availability Statement:** All relevant data are within the paper and its Supporting Information files.

## Abstract

To investigate which cytokines, chemokines and growth factors are involved in the immuno-pathogenesis of idiopathic uveitis, and whether cytokine profiles are associated with. Serum and aqueous humor (AH) samples of 75 patients with idiopathic uveitis were analyzed by multiplex immunoassay. Infectious controls consisted of 16 patients with ocular toxoplasmosis all confirmed by intraocular fluid analyses. Noninfectious controls consisted of 7 patients with Behçet disease related uveitis and 15 patients with sarcoidosis related uveitis. The control group consisted of AH and serum samples from 47 noninflammatory control patients with age-related cataract. In each sample, 27 immune mediators ± IL-21 and IL-23 were measured. In idiopathic uveitis, 13 of the 29 mediators, including most proinflammatory and vascular mediators such as IL-6, IL-8, IL-12, G-CSF, GM-CSF, MCP-1, IP-10, TNF-α and VEGF, were significantly elevated in the aqueous humor when compared to all controls. Moreover, IL-17, IP-10, and IL-21, were significantly elevated in the serum when compared to all controls. We clustered 4 subgroups of idiopathic uveitis using a statistical analysis of hierarchical unsupervised classification, characterized by the order of magnitude of concentrations of intraocular cytokines. The pathogenesis of idiopathic uveitis is characterized by the presence of predominantly proinflammatory cytokines and chemokines and vascular endothelial growth factor with high expression levels as compared to other causes of uveitis. There are indications for obvious Th-1/ IL21-Th17 pathways but also IL9-Th9 and increased IFN-γ-inducing cytokine (IL12) and IFN-γ-inducible CXC chemokine (IP-10). The combined data suggest that immune mediator expression is different among idiopathic uveitis. This study suggests various clusters among the idiopathic uveitis group rather than one specific uveitis entity.

**Funding:** This work was supported by NIH CORE Grant P30 EY08098 to the Department of Ophthalmology, the Eye and Ear Foundation of Pittsburgh, and from an unrestricted grant from Research to Prevent Blindness, New York, NY Funding by Association Inflam'Œil. The funders had no role in study design, data collection and analysis, decision to publish, or preparation of the manuscript.

**Competing interests:** The authors have declared that no competing interests exist.

## Introduction

Uveitis refers to a variety of clinical presentations with different phenotypes. Idiopathic from unknown etiology (or idiopathic uveitis) is reported for 36% of cases [1, 2], and 10% leads to blindness in developed countries. Some idiopathic uveitis could be autoimmune or infectious uveitis undiagnosed. There is still no gold standard for the diagnosis of these ocular inflammatory diseases. The idiopathic character is a diagnosis of exclusion when the clinical, radiological and biological work-up are noncontributive and the ophthalmological examination is nonpathognomonic for a particular entity. However, the question remains: what is the initial cause of those inflammatory process in the eye? Moreover, familial cases of intermediate idiopathic uveitis and other autoimmune diseases suggest that genetic variants and/or a single environmental agent are probably the cause of auto-immune diseases. Indeed, the hypothesis of a susceptibility to uveitis stemming from genetic determinants, as seen in other immunological diseases, has been initially suggested by their mode of hereditary transmission in certain families. One hypothesis would that an infectious agent (virus or bacteria) would activate systematically the autoreactive T lymphocytes in patients genetically predisposed. It is therefore possible to consider a microbial agent as an initiating or potentiating factor. We know that in certain cases, viral infections even eradicated, may have introduced immune responses, propagate these responses by using molecular mimics.

One means by which microbial agents can play a role is by their adjuvant effect, for example, in shifting the balance of the immune responses which are normally controlled by the inhibitory regulator mechanisms, toward mechanisms that predispose patients to developing one of these illnesses.

Moreover, we know very little about the immune mechanisms involved in uveitis and in particular in the idiopathic ones.

Research on the subject is limited due to the difficulty of obtaining histological samples from inflamed eyes in humans. Animal models permit the exploration of these mechanisms *in vivo* but are rarely relevant. Studies in mice show that effector cells Th1 and Th17 can independently induce tissue changes in uveitis models [3]. The eye is relatively protected from the immune system by the blood retinal barrier, by the immune inhibitor environment and active tolerance mechanisms involving CD4+ regulatory T lymphocytes (regulatory T cells or Tregs) that could influence the susceptibility to developing uveitis which is the case in other immunological diseases including multiple sclerosis (MS) or rheumatoid arthritis [4, 5]. The resident retinal cells such as the Müller glia cells and those of the pigment epithelium contribute to this micro environment by the production of cytokines. The level of these cytokines determines their diverse susceptibility to induce uveitis [6, 7].

The study of the immune mechanisms in idiopathic uveitis could answer this question.

By means of collecting aqueous humor (AH) samples we have direct access to the intra-ocular compartment, and an assay of the mediators of inflammation enabling the analysis of this inflammation at the site of activity.

The aim of this study was to identify which cytokine, chemokines and growth factors are deregulated in idiopathic uveitis and whether specific cytokines profiles are associated with clinical manifestations. To this end, cytokines, chemokines and growth factors profiles in the AH and serum were determined by multiplex immunoassay (Luminex®) technology.

## Patients and methods

### Ethics statement and subjects

This study was conducted in the Quinze-Vingts National Ophthalmologic Eye Center, Paris, France between January 2014 and May 2016. The French institutional review boards/Ethics

**Table 1. Total number of paired AH and serum samples analyzed.**

| | Biological media | | | | |
|---|---|---|---|---|---|
| | AH | | | serum | |
| | total number of samples (*n*) | number of samples analyzed (*n*) | Degree of inflammation in anterior segment & in vitreous (haze) (median) [min; max] | total number of samples (*n*) | number of samples analyzed (*n*) |
| **Patients groups** | | | | | |
| Noninflammatory controls (age-related cataract) | 36 | • 27 cytokines (36) <br> • IL-21 & IL-23 (7) | 0 <br> 0 | 47 | • 27 cytokines (47) <br> • IL-21 & IL-23 (16) |
| uveitis related to Behçet disease | 5 | • 27 cytokines (5) <br> • IL-21 & IL-23 (1) | 0 [0;1] <br> 0.5 [0;1] | 7 | • 27 cytokines (7) <br> • - IL-21 & IL-23 (6) |
| sarcoidosis | 15 | • 27 cytokines (15) <br> • IL-21 & IL-23 (3) | 1 [0;2] <br> 1 [0;3] | 12 | • 27 cytokines (12) <br> • - IL-21 & IL-23 (14) |
| toxoplasmosis | 16 | • 27 cytokines (11) <br> • IL-21 & IL-23 (3) | 1 [0;2] <br> 1 [0;2] | 16 | • 27 cytokines (16) <br> • - IL-21 & IL-23 (14) |
| idiopathic uveitis | 75 | • 27 cytokines (69) <br> • IL-21 & IL-23 (58) | 0 [0;3] <br> 1 [0;4] | 75 | • 27 cytokines (63) <br> • - IL-21 & IL-23 (68) |

AH: aqueous humor; P values found by comparing grades of anterior chamber inflammation between idiopathic uveitis versus uveitis related to sarcoidosis, Behçet disease, and toxoplasmosis (P <0.001; P = 0.137 ; P <0.0001, respectively). P values found by comparing grades of vitreous haze between idiopathic uveitis versus uveitis related to sarcoidosis, Behçet, and toxoplasmosis (P = 0.6066 ; P = 0.3543 ; P = 0.116, respectively).

Committee approval was obtained (registration number 13934 at CPP-Ile-de-France, St-Antoine Hospital, Paris, France). Written informed consents to participate to the study were obtained from all participants with none of the patients objecting to the use of the remnants of their samples for further research. This study was conducted according to the tenets of the Declaration of Helsinki.

Results of 27 cytokines, chimiokines and growth factors from paired AH and sera ($n = 69$ and 63, respectively) from 75 patients diagnosed with idiopathic uveitis were compared to mediator profiles detected in paired AH and serum samples of 47 control patients from patients with age-related cataract ($n = 47$ and $n = 36$, respectively). Results of both cytokines, IL-21 and IL-23, from paired AH and sera ($n = 68$ and $n = 58$, respectively) from the patients with idiopathic uveitis were compared to the ones from controls age-related cataract ($n = 16$ and 7, respectively). Additional information on the patient groups were presented in Table 1.

Paired sera and AH samples obtained from patients with uveitis related to toxoplasmosis (TU) ($n = 16$), uveitis related to Behçet disease ($n = 7$) and ocular sarcoidosis (N = 15) served as disease controls. The same exclusion criteria than for the uveitis group applied to the healthy donors and they did not have any clinical symptoms of uveitis. Exclusion criteria was glaucomatous and diabetes because elevated IL-6 and IL-8 levels were also reported in glaucomatous aqueous with elevated IOP and diabetes [8, 9].

Patients were identified and referred for enrolment by ophthalmologists after careful examination and evaluation by Internal Medicine specialists. The diagnosis of uveitis at the moment of sample withdrawal was based on an ophthalmic examination consisting of visual acuity recordings (Snellen scale), slit-lamp examination, grading of anterior chamber cells and vitreous haze using the SUN grading system [10], intraocular pressure measurement, and dilated fundus examination with indirect ophthalmoscopy. Ancillary testing such as fluorescein angiography and optical coherence tomography were performed when necessary. The classification of uveitis used was the anatomical classification (the International Uveitis Study Group

(IUSG) [10]. The criteria for idiopathic (or unknow etiologies) were investigations oriented by the anatomic characteristics of uveitis: negative serologic screening for syphilis, normal serum angiotensin-converting enzyme, and interferon-gamma release, normal chest computed tomography. Our group has published a standardized strategy that we use in routine for the etiologic diagnosis of uveitis with first (CBC, ESR, CRP, quantiferon, syphilis serology, chest radiograph), second (ACE, antinuclear antibodies, complement, HLA B27 etc. . .) and third steps investigations based on the clinical type of uveitis and clinical and medical history findings. A cerebral magnetic resonance imaging and anterior chamber tap with interleukin-10 analysis and cytology, Herpes viridea (HSV, VZV, CPV) PCR and/or Goldmann coefficient are part of the second/ third steps investigations for chronic intermediate, posterior and panuveitis or when severe and/or corticoresistant uveitis [11]. We excluded patients based any past history of systemic inflammation, auto-immune disease, concomitant anti-inflammatory treatment, immunosuppressed state or systemic antibiotics or immunomodulatory therapy within 4 weeks before inclusion.

In this study, paired AH and serum samples of 75 patients with idiopathic uveitis were included.

-The 47 patients who underwent cataract extraction (27 women and 20 men; median age 71 years [30–100 years]) and served as a control group had no history of uveitis. Sera and AH samples were collected prior to cataract extraction. The baseline level of cytokines/ chemokines in AH was determined using samples from the control group.

-For control group consistent with TU and serving as infectious disease controls, the diagnosis of TU was confirmed by real-time PCR detection of *Toxoplasma gondii* DNA or a Goldmann-Witmer test to prove intraocular specific antibody synthesis. Patients who were immunocompromised, suffered from other ocular infections, or receiving local or systemic anti-Toxoplasma treatment for active uveitis, were excluded.

With regard to rheumatologic and ophthalmic disorders, we used the the International Study Group criteria for Behçet disease [12], and international criteria for the diagnosis of ocular sarcoidosis [13].

## Biological analysis

Paired samples of AH and serum were obtained from each subject at the time of clinical diagnosis for laboratory analysis. AH samples (100–150 μL) were collected through anterior chamber paracentesis and stored, along with serum samples, at −80˚C until analysis.

In each sample, 27 immune mediators were analyzed: 4 anti-inflammatory cytokines (interleukin IL-1 receptor antagonist [IL-1Rα], [IL]-4, IL5, IL-10, and IL-13); 12 proinflammatory mediators (cytokines IL-1β, IL-2, IL-6, IL-12p70, IL-17, interferon-γ [IFN-γ], tumor necrosis factor-α [TNF-α], and chemokines IL-8 [CXCL8], interferon-inducible 10-kDa protein [IP-10; CXCL10], monocyte chemotactic protein-1 [MCP-1; CCL2], macrophage inflammatory protein-1α [MIP1α; CCL3]; and -1β [MIP-1β; CCL4]; 3 additional mediators (cytokines IL-15 and macrophage migration inhibitory factor [MIF], and chemokines RANTES [regulated on activation, normal T-cell expressed and secreted; CCL5] and Eotaxin [CCL11]); granulocyte-macrophage colony-stimulating factor [G-CSF], granulocyte-macrophage colony-stimulating factor [GM-CSF], 4 growth factors [hematopoietic growth factor [IL-7], Fibroblast growth factor [FGF Basic], Platelet-derived growth factor [PDGF-BB], vascular endothelial growth factor [VEGF]].

AH and serum samples were analyzed by multiplex immunoassay (Luminex®). The assay plate layout consisted in a standard series in duplicate (1 to 32 000 pg/mL), four blankwells and 10μL duplicates of AH samples, diluted to 50 μL with BioPlex Human serum diluent. Quantitative determination was performed using an Invitrogen Human Cytokine 27-Plex

Panel. The 27 plex was enriched with and one separate Invitrogen Human Cytokine 2-Plex Panel for IL-21, IL-23 and in accordance with the manufacturer's protocol (BioRad®).

### Statistical analysis

Data were presented as median and range (min, max). Non-parametric Kruskal-Wallis and Fisher's exact tests were performed to compare continuous variables, as appropriate. P values less than 0.05 were considered significant. The statistical analyses were performed using GraphPad Prism version 8.0.1, Graph Pad Software, Inc, San Diego, CA.

The comparaison of dosage of different cytokines, chemokines and growth factors between idiopathic uveitis and various controls was done using a non-parametric test of Kruskal-Wallis.

The representation of cytokine distributions (boxplots) was performed according to pathology groups, with comparisons between controls vs other pathologies (Behcet, sarcoidosis and toxoplasmosis) with a correction of P-values with the method of Bonferroni (to avoid alpha risk inflation due to multiple comparisons).

The classification of cytokines in idiopathic uveitis was done by selecting only the uveitis significantly different from those of the controls. The method used is the hierarchical unsupervised classification after focusing and reducing the data (subtract the mean and divide by standard deviation) in order to report the cytokines in the same unit (around 0).

A distribution of clinical data according to the groups identified in the classification was presented. No comparison tests were conducted because of the exploratory nature of the analysis as well as the low number per group.

### Results chemokines, cytokines and growth factors in the serum

Patients with idiopathic uveitis exhibited higher levels of IP-10, IL-17, and IL-21 than serum samples of cataract patients.

Specifically, median levels of chemokines and cytokines IP-10, IL-17, and IL-21 were significantly elevated in the serum of patients with idiopathic uveitis as compared with nonflammatory controls: 671 pg/mL [157–3063] vs 526 pg/mL, for IP-10 ; 173 pg/mL [39–500] vs 49 pg/mL for IL-17 and 28 pg/mL [0–382] vs 0 pg/mL for IL-21 (p = 0.0032, p< 0,0001, p = 0.0007, respectively) (Fig 1). Median levels of IL-23 were decreased in the serum of patients with idiopathic uveitis as compared with noninflammatory controls: 11 pg/mL [0–187] vs 6 mg/mL [0–22] (p< 0.0001).

However, median levels of the following mediators in serum in patients with idiopathic uveitis were not significantly different as compared with controls: proinflammatory cytokines and chemokines IL-1β, IL-6, IFN-γ, TNF-α, MCP-1, G-CSF, MIP-1α, and MIP-1β; anti-inflammatory cytokines IL-10, IL1-Rα, and the angiogenic growth factor VEGF.

Some patients with idiopathic uveitis had increased levels of chemokines, cytokines and growth factors as compared with the cut-off defined in control patients (mean + 3 standard deviation): IL-23 in 6 patients, IL-7 in 7 patients, IL-1β and PDGF-BB in 5 patients; IL-6 in 4 patients ; IL-1Rα in 3 patients; IL-2, IL-4, IL-10, IL-12, GM-CSF, VEGF in 2 patients and IL-15, G-CSF, IFN-γ, MIP-1α, MIP-1β, RANTES, TNF-α in one patient ; nonetheless, differences between median levels found in the samples from patients with idiopathic uveitis as compared with the controls' were not significant for those mediators (Fig 2).

### Results chemokines, cytokines and growth factors in the AH (Table 2 & Figs 3–7)

Patients with uveitis exhibited higher levels of IL-5, IL-6, IL-8, IL-12, IP-10, MCP-1, G-CSF, GM-CSF, MIP-1α, MIP-1β, TNF-α and VEGF than intraocular fluid samples of cataract patients, whilst IL-4 was lower.

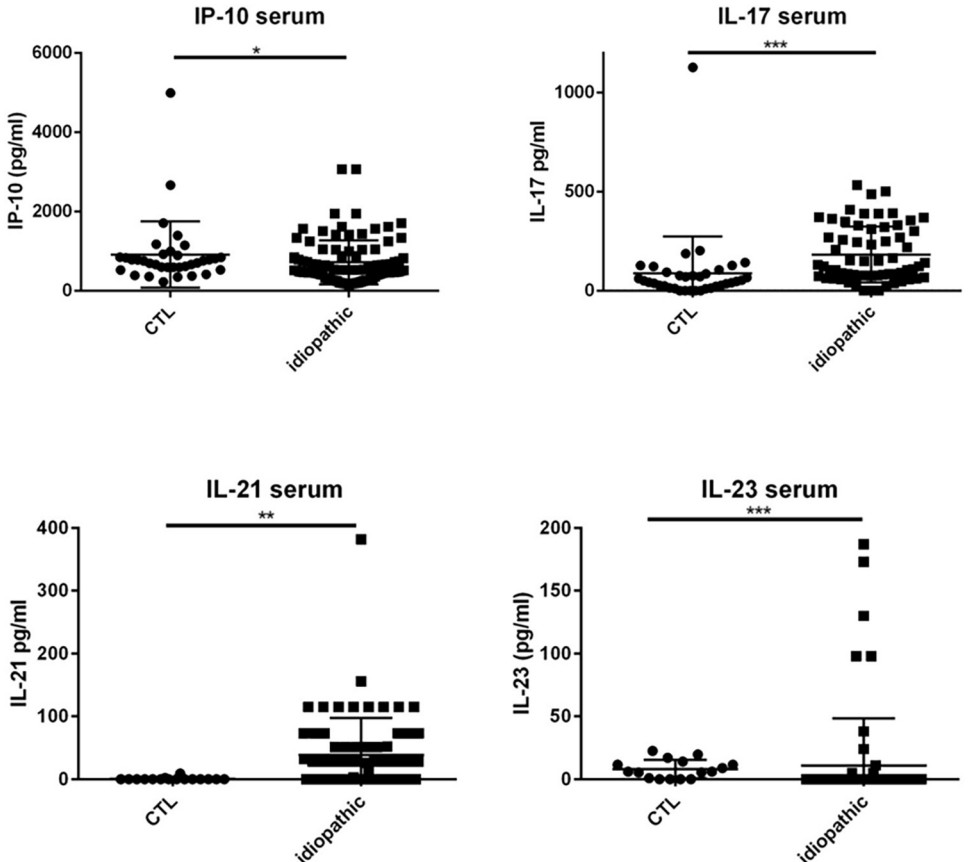

**Fig 1. Dot plots of immune mediators, IL-17 and IP-10 in the serum.** IL-17 and IP-10 in the serum are increased in patients with idiopathic uveitis (n = 63) compared with serum of noninflammatory controls patients (cataract age-related) (CTL) (n = 47). Medians levels for IL-21 and IL-23 in the serum: elevated for IL-21 and decreased for IL-23 in patients with idiopathic uveitis (N = 68) as compared with noninflammatory controls patients (cataract age-related) (CTL) (N = 16). * = P<0.05 ; ** = P<0.01; *** = P<0.001.

Specifically, median levels of chemokines, cytokines, and growth factors were significantly increased in the AH of idiopathic uveitis as compared with noninflammatory controls: IL-1Rα was 50.92 pg/mL [0–126.9] vs 0.83 pg/mL, IL-5 was 0 pg/mL [0–5.52] vs 0 pg/mL; IL-6 was 81.73 pg/mL [8.82–511.2] vs 6.64 pg/mL, IL-8 was 22.23 pg/mL [2.12–87.86] vs 2.76, l'IL-9 was 2.85 pg/mL [0–8.8] vs 0 pg/mL, IL-12 was 11.13 pg/mL [5.67–20,49] vs 3.30; Eotaxine was 6.29 pg/mL [0–30.61] vs 0 pg/mL ; IP-10 was 4442 pg/mL [462.8–17790] vs 284.74; G-CSF was 9.98 pg/mL [1.47–113.3] vs 0.64 pg/mL, MCP-1 was 125.2 pg/mL [46.24–315.8] vs 59; GM-CSF was 3.5 pg/mL [0–100.40] vs 0 pg/mL; MIP-1α was 1.21 pg/mL [0–2.66] vs 0 pg/mL, MIP-1β was 27.2 pg/mL [11.16–47.61] vs 0 pg/ml, le TNF-α with 0 pg/mL [0–6.3] vs 0 pg/mL and VEGF with 79.19 pg/mL [26.84–160.6] vs 0 pg/mL.

Conversely, mean levels of IL-4, IL-7, IL-15 and PDGF-BB were significantly decreased in AH of idiopathic uveitis as compared with noninflammatory controls.

We found some patients with idiopathic uveitis and higher levels of IL-1β, IFN-γ and IL-23 in AH compared with noninflammatoy controls, although their median of concentration was not significantly increased compared with controls (Table 2).

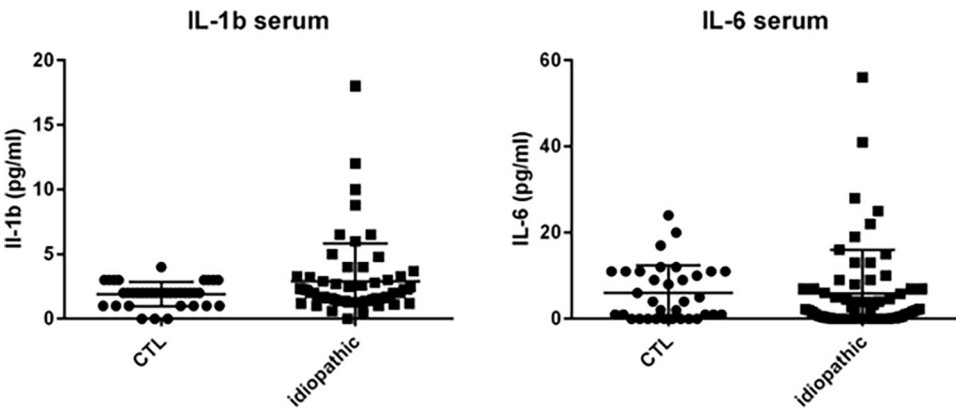

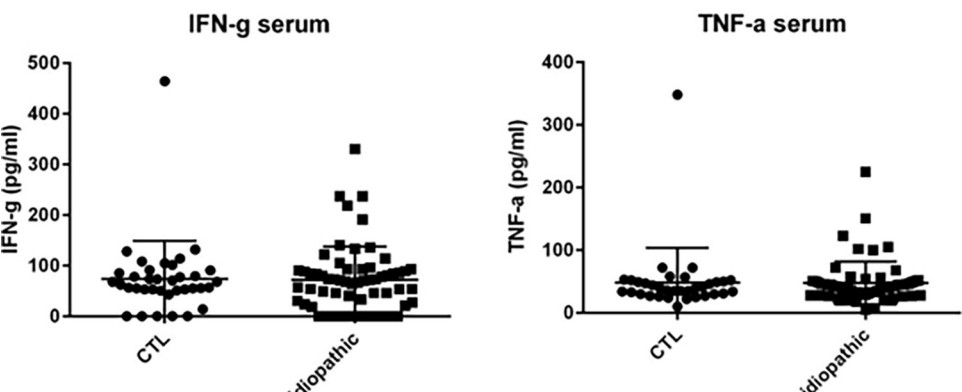

**Fig 2. Dot plots of immune mediators, IL-1β, IL-6, IFN-γ and TNF-α.** IL-1β, IL-6, IFN-γ and TNF-α in serum of noninflammatory controls patients (cataract age-related) (CTL) (N = 36) as compared with immune mediators in the serum of patients with idiopathic uveitis (n = 51) but differences in both groups are not significant. Note that few isolated patients with idiopathic uveitis had cytokines increased in serum samples. * = P<0.05 ; ** = P<0.01; *** = P<0.001.

## Uveitis-associated cytokines/chemokines/ growth factors cluster represent various inflammatory ocular biology pathways

We clustered 4 subgroups of idiopathic uveitis using a statistical analysis of hierarchical unsupervised classification, characterized by the order of magnitude of concentrations of intraocular mediators (cytokines/ chemokines/ growth factors) (Fig 8).

The 1st cluster (in pink in Fig 8) presents AH mediators concentrations slightly increased (although significantly different from those found in the control group); the 2nd cluster (in green in Fig 8) presents AH mediators concentrations mildly increased; the 3rd cluster (in blue in Fig 8) presents mediators concentrations moderately increased and concentrations of mediators IL-7, IL-15 and PDGF-BB quite similar of those found in noninflammatory controls; and the 4th cluster (in black in Fig 8) presents a higher increase of levels of mediators.

**Table 2. Concentrations (pg/ml) of immune mediator expression in aqueous humor (AH) from 75 samples of patients with idiopathic uveitis and 36 samples of noninflammatory controls patients (age-related cataract).** *n* = patients number.

| Variable | Groups | | P* |
|---|---|---|---|
| | Idiopathic uveitis, *n* = 75 | Noninflammatory controls (age-related cataract without uveitis), *n* = 42 | |
| IL-1β | 0 [0–0.46] | 0 [0–0.43] | 0.1033 |
| IL-1Rα | 50.92 [0–126.9] | 0.83 [0–61.69] | **0.0342** |
| IL-2 | 0 [0–0] | 0 [0–0] | 0.8926 |
| IL-4 | 0 [0–0] | 0 [0–0,4] | **0.0439** |
| IL-5 | 0 [0–5.52] | 0 [0–0] | **<0.0001** |
| IL-6 | 81.73 [8.82–511.2] | 6.64 [2.3–40.96] | **0.0079** |
| IL-7 | 0 [0–18.72] | 7.63 [1.89–33.11] | 0.1194 |
| IL-8 | 22.23 [2.12–87.86] | 2.76 [1.62–6.36] | **0.0013** |
| IL-9 | 2.85 [0–8.8] | 0 [0–0.34] | **0.0004** |
| IL-10 | 0 [0–10.18] | 0.35 [0–2.16] | 0.1115 |
| IL-12 | 11.13 [5.67–20.49] | 3.3 [1.07–6.57] | **<0.0001** |
| IL-13 | 0.46 [0–3.9] | 0.1 [0–1.08] | 0.2153 |
| IL-15 | 0 [0–0] | 0 [0–8.56] | **0.0042** |
| IL-17 | 0 [0–9.96] | 0 [0–115.0] | **0.0204** |
| IL-21 | 0 [0–26.09] | 0 [0–0] | 0.0504 |
| IL-23 | 0 [0–4.92] | 0 [0–0] | 0.0338 |
| Eotaxin | 6.29 [0–30.61] | 0 [0–0] | **<0.0001** |
| FGF-basic | 0 [0–0] | 0 [0–0] | 0.1238 |
| G-CSF | 9.98 [1.47–113.3] | 0.64 [0–1.89] | **<0.0001** |
| GM-CSF | 0 [0–100.4] | 0 [0–2.39] | 0.1270 |
| IFN-γ | 0 [0–0] | 0 [0–0] | 0.1119 |
| IP-10 | 4442 [462.8–17790] | 284.7 [134.8–484.6] | **<0.0001** |
| MCP-1 | 125.2 [46.24–315.8] | 59 [4.11–95.26] | **<0.0001** |
| MIP-1α | 1.21 [0–2.66] | 0 [0–0] | **0.00001** |
| PDGF-bb | 0 [0–0] | 1.69 [0–13.46] | **0.0728** |
| MIP-1β | 27.2 [11.16–47.61] | 0 [0–7.88] | **<0.0001** |
| RANTES | 0 [0–0] | 0 [0–0] | 0.1656 |
| TNF-α | 0 [0–6.3] | 0 [0–0] | **0.0006** |
| VEGF | 79.19 [26.84–160.6] | 0 [0–24.37] | **<0.0001** |

*Significant P values are noted in right column of the Table. Statistical analysis was done with non parametric Kruskal-Wallis and Fisher's exact tests for the comparison of dosage of different cytokines between idiopathic uveitis and controls in aqueous humor (medianes of concentrations). A P value < 0.05 was considered significant.

Three patients' samples were excluded from the biostatistical analysis (Fig 8) because they presented levels of mediators to high compared with the others. Those 3 patients had idiopathic uveitis of panuveitis type (two patients had relapsing uveitis of that kind).

Clinical features of these 4 clusters of patients are presented in Table 3.

We compared the median AH concentrations of cytokines and chemokines between noninflammatory controls and patients with uveitis related to Behçet disease, sarcoidosis, TU and idiopathic uveitis (Fig 9) for those 4 mediators significantly elevated in idiopathic uveitis as compared as noninflammatory controls: IL-6, TNF-α, IL-12 and IP-10. IL-6, TNF-α and P-10 were found significantly elevated in the AH of patients with uveitis related to Behçet disease, sarcoidosis and TU as compared with noninflammatory controls. IL-12 was found elevated in all uveitis causes as compared as noninflammatory controls except in TU.

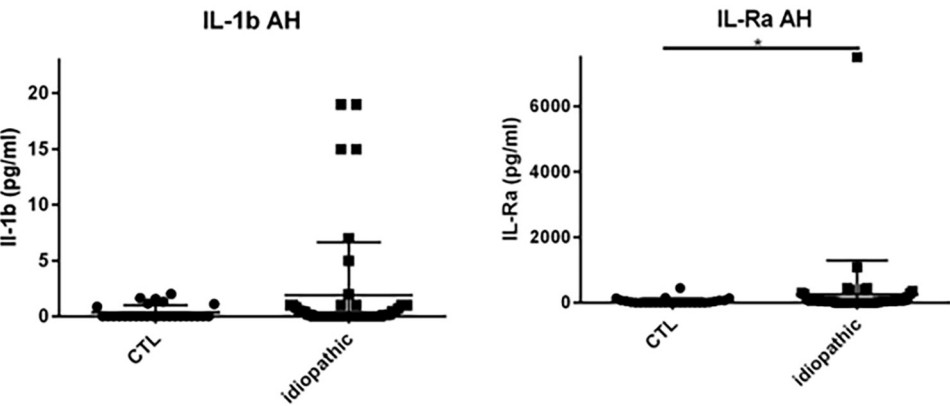

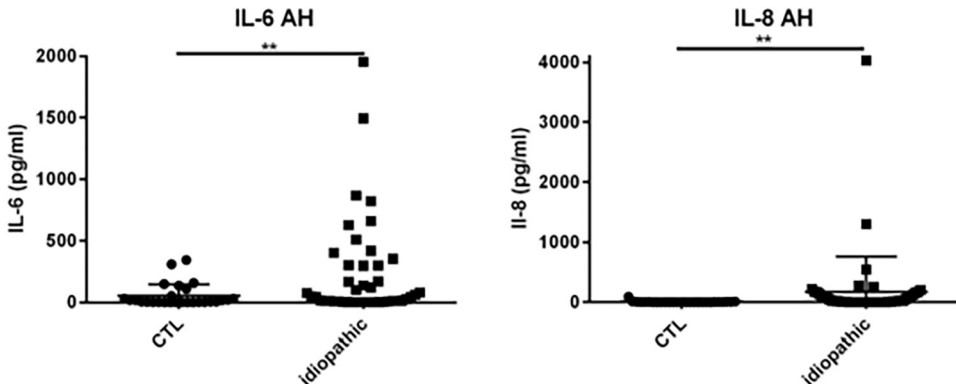

**Fig 3. Dot plots of immune mediators: IL-1β, IL1-Rα, IL-6, IL-8, in aqueous humor of patients with idiopathic uveitis (N = 64) as compared with immune mediators in the aqueous humor of noninflammatory controls patients (cataract age-related) (N = 36).** Note that few isolated patients with idiopathic uveitis had an immune mediator elevated. * = P<0.05 ; ** = P<0.01; *** = P<0.001.

We further compared the same cytokines and chemokines (IL-6, TNF-α, IL-12 et IP-10), between noninflammatory controls with uveitis related to sarcoidosis, TU and idiopathic uveitis with the exclusion of Behçet disease related uveitis, due to to disparate orders of values (Fig 10): AH contained much higher levels of those mediators in Behçet disease as compared as other causes of uveitis (Fig 10).

We compared the 4 subgroups of idiopathic uveitis defined by the statistical method shown in Fig 8.

The first group had relatively weak cytokine values, although significantly different from those of the non-inflammatory control ('idiop 1'); A second group presented moderate high cytokine values ('idiop 2'); A third group presented moderate high cytokine values (IL-6, TNF-α, IP-10) except for the IL-12, which is very high ('idiop 3'); and finally, a fourth group had very high cytokines and chemokines values ('idiop 4').

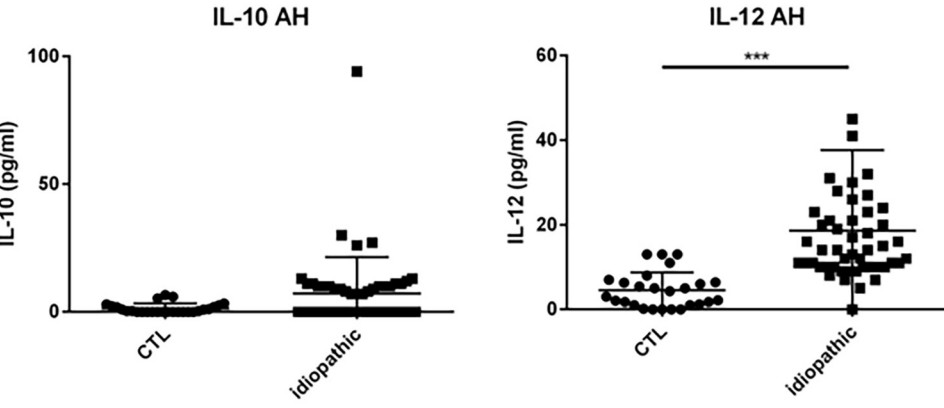

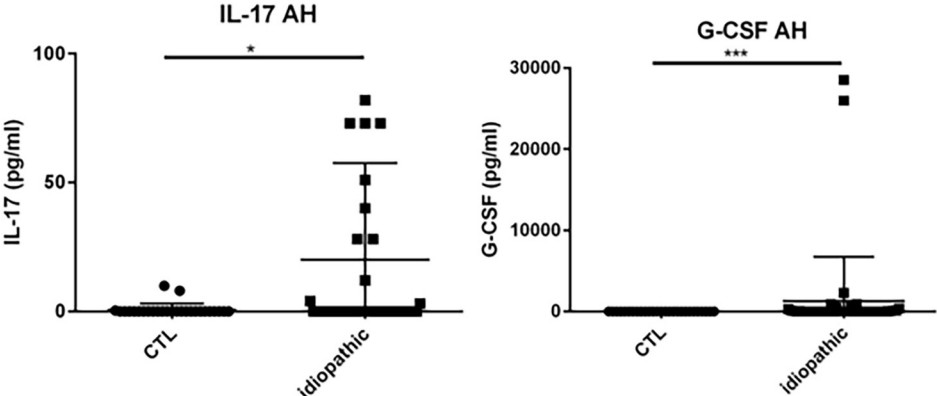

**Fig 4. Dot plots of immune mediators: IL-10, IL-12, IL-17, G-CSF, in aqueous humor of patients with idiopathic uveitis (N = 64) as compared with immune mediators in the aqueous humor of noninflammatory controls patients (cataract age-related) (N = 36).** Note that few isolated patients with idiopathic uveitis had an immune mediator elevated. * = P<0.05 ; ** = P<0.01; *** = P<0.001.

## Discussion

Idiopathic uveitis is the predominant diagnosis in most series from tertiary clinics. Idiopathic uveitis is one of the most challenging inflammatory eye disease because of the lack of understanding of the mechanism governing the disease. This study describes the Th17/Th1/Th2 cytokine, chemokine and growth factor profile in aqueous humor samples and serum from eyes with idiopathic uveitis, determined with a multiplex-based technology.

The analysis of AH samplings from patients with idiopathic uveitis confirmed elevated levels of cytokines and chemokines mostly of inflammatory type as compared with noninflammatory controls. In that, our study is in accordance with previous studies [14, 15].

In particular, idiopathic uveitis seems being associated with higher AH levels of the following immune mediators: IL-1Rα, IL-6, IL-8, IL-12, IL-17, IP-10, MIP-1α, MIP-1β, MCP-1, G-CSF and TNF-α. We also found other mediators elevated, IL-5 and IL-9.

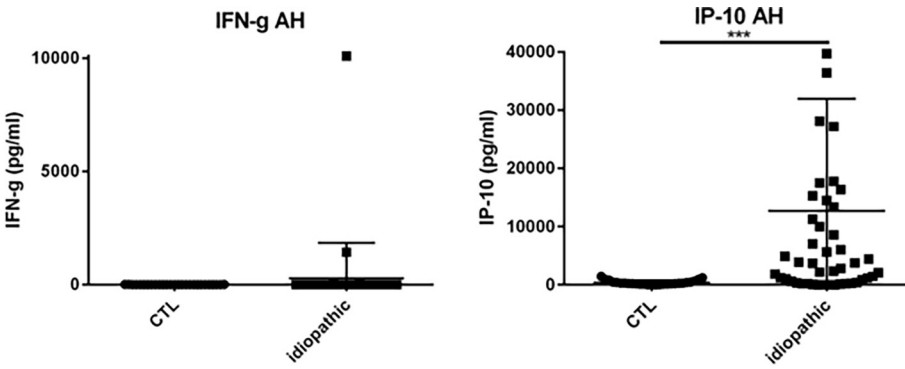

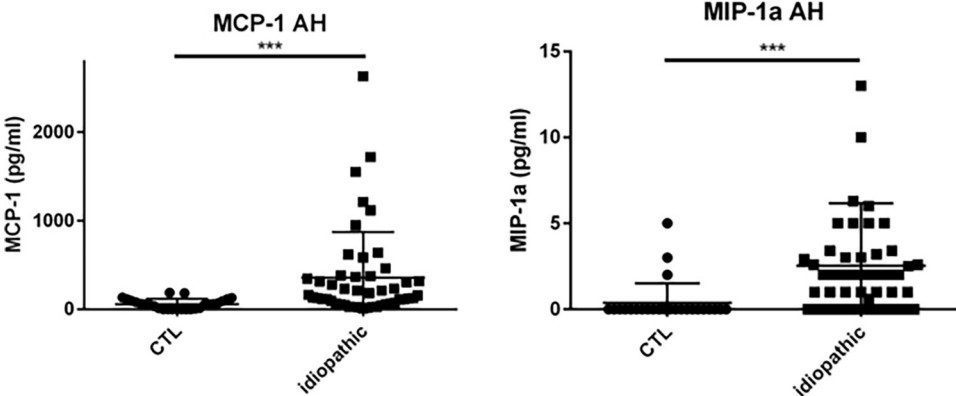

**Fig 5. Dot plots of immune mediators: IFN-γ, IP-10, MCP-1, MIP-1α, in aqueous humor of patients with idiopathic uveitis (N = 64) as compared with immune mediators in the aqueous humor of noninflammatory controls patients (cataract age-related) (N = 36).** Note that few isolated patients with idiopathic uveitis had an immune mediator elevated. * = P<0.05 ; ** = P<0.01; *** = P<0.001.

To our knowledge, our study is unique in that we focused specifically on idiopathic uveitis and their immune mediators. Previous studies have considered cytokines in patients with idiopathic uveitis localized in the intermediate eye segment, however without distinguishing idiopathic uveitis from multiple sclerosis, sarcoidosis or Lyme borreliosis. In particular, higher AH levels have been shown for IL-1β, IL-6, IL-8, IL-10, IL-12(p70), IFNγ and CCL2/MCP-1 as compared with noninflammatory controls [15].

First, our study described the Th1 cytokines, i.e., TNF-α, IFN-γ-inducing cytokine (IL-12) and IFN-γ-inducible CXC chemokine (IP-10).

Our data show that increased levels in HA of mediators involved in the production and the activity of IFN-γ may play an important immunopathogenic role in idiopathic uveitis.

We confirmed common knowledge that some patients with idiopathic uveitis have increased AH level of IFN-γ. In our study, median of concentrations of INF-γ was not significantly elevated in the whole cohort of patients with idiopathic uveitis but three patients had AH levels higher than the cut-off as defined in noninflammatory controls (cut-off = mean + 3 standard deviations). Previous studies have shown significantly AH elevated IFN-γ in patients

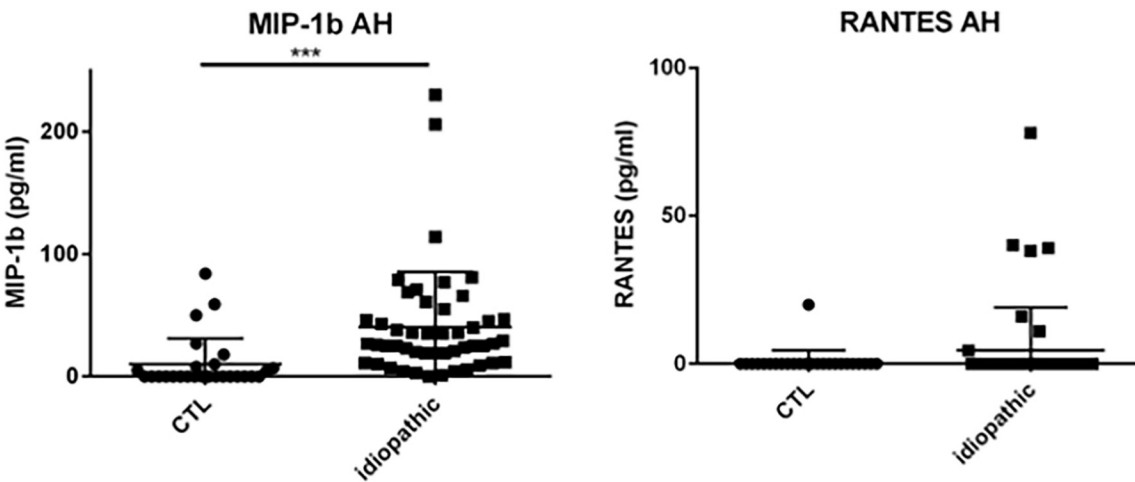

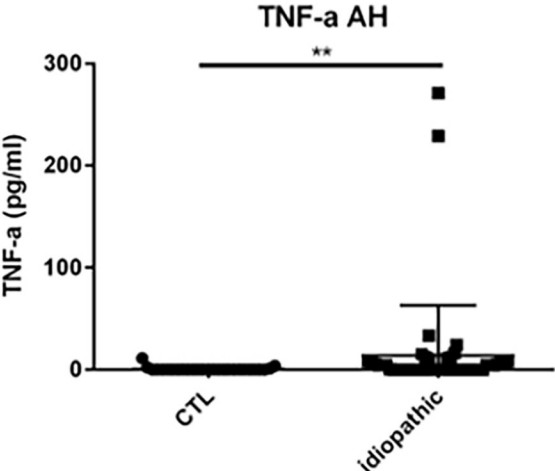

**Fig 6. Dot plots of immune mediators: MIP-1β, RANTES, TNF-α, in aqueous humor of patients with idiopathic uveitis (N = 64) as compared with immune mediators in the aqueous humor of noninflammatory controls patients (cataract age-related) (N = 36).** Note that few isolated patients with idiopathic uveitis had an immune mediator elevated. * = P<0.05 ; ** = P<0.01; *** = P<0.001.

with idiopathic anterior uveitis of anterior and intermediate anatomical type [14, 16–18]. In herpetic viral etiology, mean level of IFN-γ is higher than in other viral uveitis and in noninfectious uveitis [19, 20].

In our study, IL-12 was the cytokine most commonly elevated in patients with idiopathic uveitis and its median level was also elevated in the AH. IL-12 is an immunoregulatory cytokine which plays a key role in the polarization of the naïve Th cells and influences IFN-γ production [21–23]. Our results regarding IL-12 confirms the immune mechanism Th1 serving a preponderant role in human uveitis [22]. Although intraocular levels of IL-12 have been

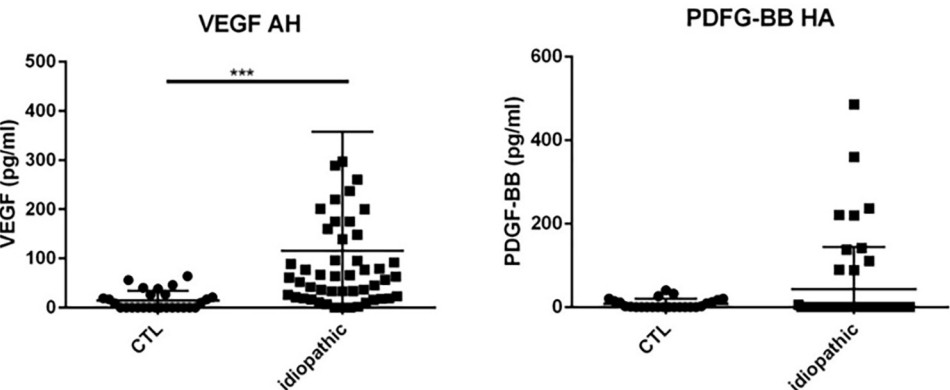

**Fig 7. Dot plots of immune mediators: VEGF and PDGF-BB, in aqueous humor of patients with idiopathic uveitis (N = 64) as compared with immune mediators in the aqueous humor of noninflammatory controls patients (cataract age-related) (N = 36).** Note that few isolated patients with idiopathic uveitis had an immune mediator elevated. * = P<0.05 ; ** = P<0.01; *** = P<0.001.

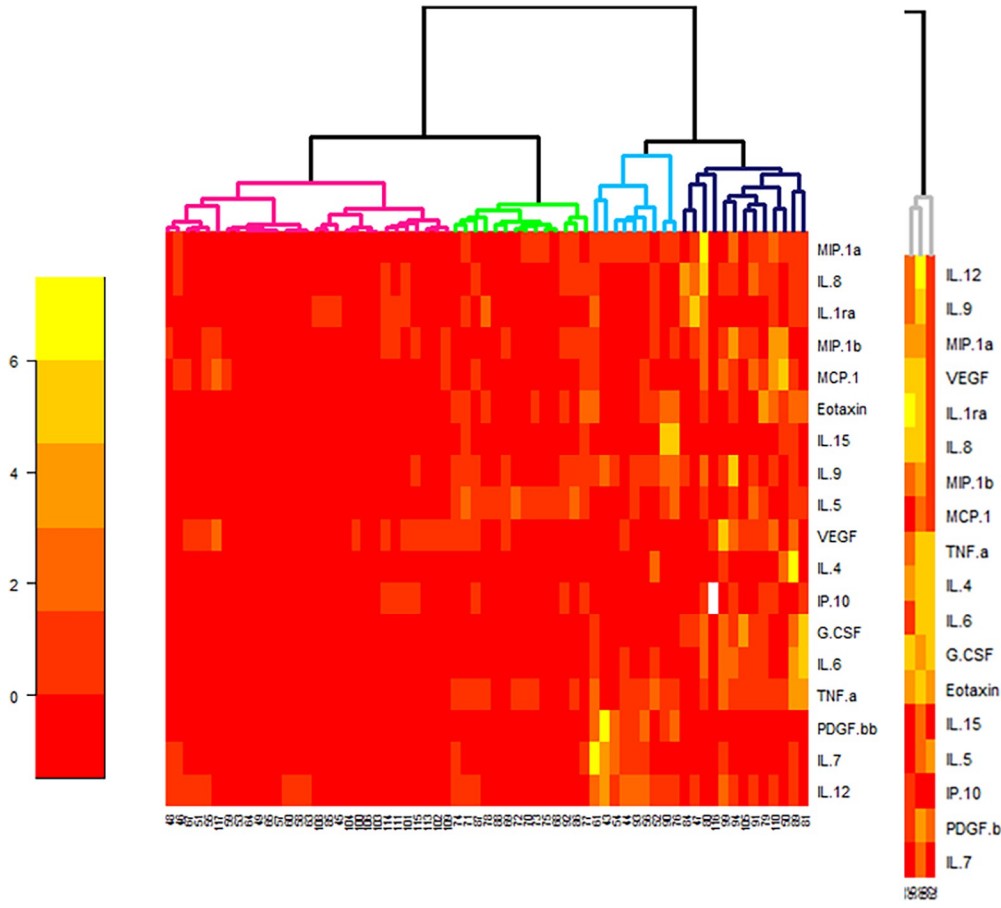

**Fig 8. Heat map analysis comparing aqueous humor cytokine concentrations in 64 the idiopathic uveitis samples and 47 non-inflammatory controls (age-related cataracts).** Only significant different cytokines in idiopathic uveitis compared with controls were considered. Cytokine concentrations are depicted as colors ranging from red to orange to yellow, indicating low, intermediate, and high concentration, respectively, relative to the mean cytokine concentration. The phenotype of the samples is indicated by the label on the right as well as by the colored bar on the top. Samples are separated in one of four clusters (pink, green, blue and black) indicated on the top.

**Table 3. Comparison between 4 sub-groups patients with idiopathic uveitis (*n* = 64\*) and their clinical parameters.**

| | | Groups | | | |
|---|---|---|---|---|---|
| variable | | 1 N = 26 | 2 N = 14 | 3 N = 8 | 4 N = 13 |
| Sex | | 13 (50%) | 6 (43%) | 5 (62%) | 9 (69%) |
| Age, median | | 38.5 [22;94] | 43.5 [26;88] | 50 [20;95] | 62 [20;87] |
| eye | OD | 7 (27%) | 6 (43%) | 4 (50%) | 3 (23%) |
| | OS | 4 (15%) | 3 (21%) | 3 (38%) | 4 (31%) |
| | OU | 15 (58%) | 5 (36%) | 1 (12%) | 6 (46%) |
| uveitis | intermediate | 5 (19%) | 7 (50%) | 3 (38%) | 3 (23%) |
| | posterior | 16 (62%) | 4 (29%) | 5 (62%) | 5 (38%) |
| | panuveitis | 5 (19%) | 3 (21%) | 0 (0%) | 5 (38%) |
| vasculitis | absent | 11 (42%) | 9 (64%) | 5 (62%) | 9 (69%) |
| | present | 15 (58%) | 5 (36%) | 3 (38%) | 4 (31%) |
| | venous | 11 (42%) | 3 (21%) | 3 (38%) | 3 (23%) |
| | arterial | 1 (4%) | 0 (0%) | 0 (0%) | 1 (8%) |
| | venous+ arterial | 3 (12%) | 2 (14%) | 0 (0%) | 0 (0%) |
| degree of inflammation in anterior segment | (median; min-max) | 0 [0;3] | 0 [0;1] | 0 [0;0] | 0.5 [0;2] |
| degree of inflammation in vitreous | vitritis (median; min-max) | 1 [0;3] | 0.75 [0;2] | 1 [0;2] | 1 [0;4] |
| choroidal granulomas | yes | 2 (8%) | 1 (7%) | 0 (0%) | 2 (15%) |
| macular edema | yes | 5 (19%) | 7 (50%) | 2 (25%) | 5 (42%) |
| papillitis | yes | 2 (8%) | 6 (43%) | 1 (12%) | 4 (31%) |
| course | acute | 11 (42%) | 3 (21%) | 5 (62%) | 5 (38%) |
| | relapsing | 7 (27%) | 3 (21%) | 1 (12%) | 5 (38%) |
| episode | chronic | 8 (31%) | 8 (57%) | 2 (25%) | 3 (28%) |

OD: right eye; OS: left eye; OU: both eyes; grading of anterior chamber cells and vitreous haze using the SUN grading system [10].

\*Three patients were excluded because of very high levels of chemokines/ cytokines (see Fig 8). Those patients had no common clinical signs except their age: one patient was 14 yo and 2 patients were respectively 66 and 78 yo. The two previous patients experienced relapsing uveitis episodes.

shown higher in patients with intermediate uveitis than in noninflammatory controls' eyes, those levels were not correlated to the activity of the disease in those previous studies [21, 24–26].

We found elevated levels of IP-10 in the AH and the serum of patients with idiopathic uveitis. IP-10 is linked to the monokines induced by IFN-γ (IFN-γ-inducible CXC chemokine) and to the IFN-inducible T cell α chimioattractant, that controls the migration and adhesion of activated T cells and NK cells [4–27]. The IP-10 expression is increased in various cells, including endothelial cells, keratinocytes, fibroblastes, astrocytes, moncytes and neutrophiles by stimulation of IFN-α, IFN-β, IFN-γ, LPS and in T cells by antigen activation [28]. IP-10 is also expressed in many Th1 mediated human diseases. IP-10 levels are correlated to the infiltration by the T cells suggesting that IP-10 plays a role in the attraction of T cells towards the sites of inflammation [29, 30]. IP-10 is also a chemoattractant for monocytes/macrophages, NK cells and dendritic cells [31, 32].

The median level of TNF-α of increased in our study representing 14 among 69 (20%) of the AH samples from the patients with idiopathic uveitis (Table 4, Supplemental data). There are discordant previous results as regards to AH levels of proinflammatory cytokine TNF-α in idiopathic uveitis. For Valentincic et al, the TNF-α levels in active idiopathic uveitis and in the anatomic intermediate uveitis type, did not seem being increased [14], conversely to another report of noninfectious uveitis [19]. TNF-α, is essential for the induction and maintenance of inflammation in the autoimmune reactions and is released by macrophages and T

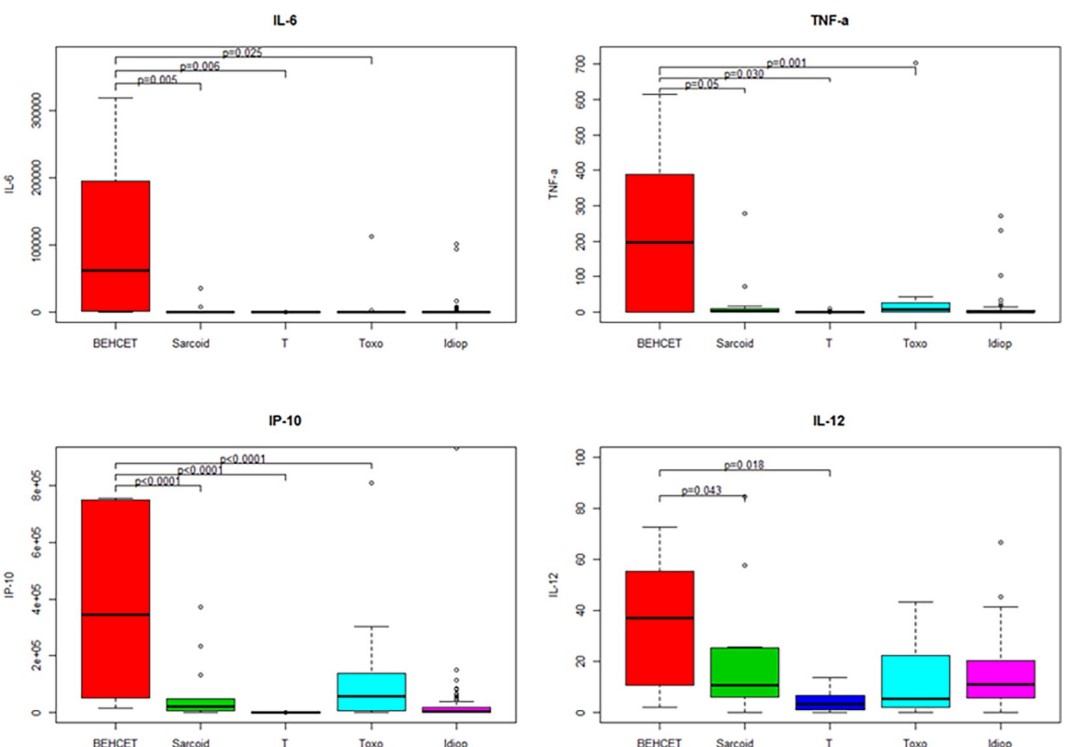

**Fig 9. Boxplots of 4 immune mediators significantly elevated in the aqueous humor of patients with idiopathic uveitis (Idiop): IL-6, TNF-α, IL-12 and IP-10, as compared to immune mediators in the aqueous humor of patients with uveitis related to Behçet disease, sarcoidosis, TU (Toxo) and noninflammatory controls (T).** Significant P values are noted in the upper part of each graph. A P value < 0.05 was considered significant.

lymphocytes during the inflammatory response. It affects the activation of leukocytes and their infiltration by upregulation of adhesions molecules and activation of macrophages. It also drives the lymphocytic Th1 response in the tissues [33].

Next, our study described the Th17 cytokines (IL-17A, IL-17F, IL-21, IL-22 and IL-23).

We found that IL-17 was significantly elevated in both serum and AH and IL-21 was elevated in the serum of patients with idiopathic uveitis. It seems that some idiopathic uveitis might be associated to a concomittant active systemic inflammation, of Th17 nature, that is herein also found in the serum samples. IL-17 elevated levels have been also described in the serum of sarcoidosis [34] and IL-17/ IL-21 in birdshot chorioretinopathy and in Vogt-Koyanagi-Harada disease [35, 36]. IL-17 is produced by a subset of CD4+ cells refered to as Th17 cells but also by T CD8+ cells [37], NK cells [38] and by γδ lymphocytes [38].

IL-17 controls the expression of cells that express the IL-17 receptor (IL-17R) with an increased secretion of IL-6, IL-8, MCP-1 and G-CSF with the induction of a chronic inflammation with monocytes and macrophages infiltrates [39]. This might correspond to the pathophysiological mechanism of idiopathic uveitis because we demonstrated elevated median levels of IL-6, de MCP-1 and G-CSF in the AH of the samples we analyzed. Of note, higher levels of MCP-1 in AH have also been previously described in idiopathic uveitis but in the anatomical type of intermediate uveitis, only [12].

Next, we studied the Th2 and Th9 cytokines (IL-4, IL-10, IL-13 et IL-9).

In 10 AH samples among the 69 that were tested in our study, IL-10 levels were found (14.5%) elevated. Although IL-10 levels were lower than IL-6 excluding the differential diagnosis of primary ocular lymphoma [40]. This is in disagreement with a previous study that

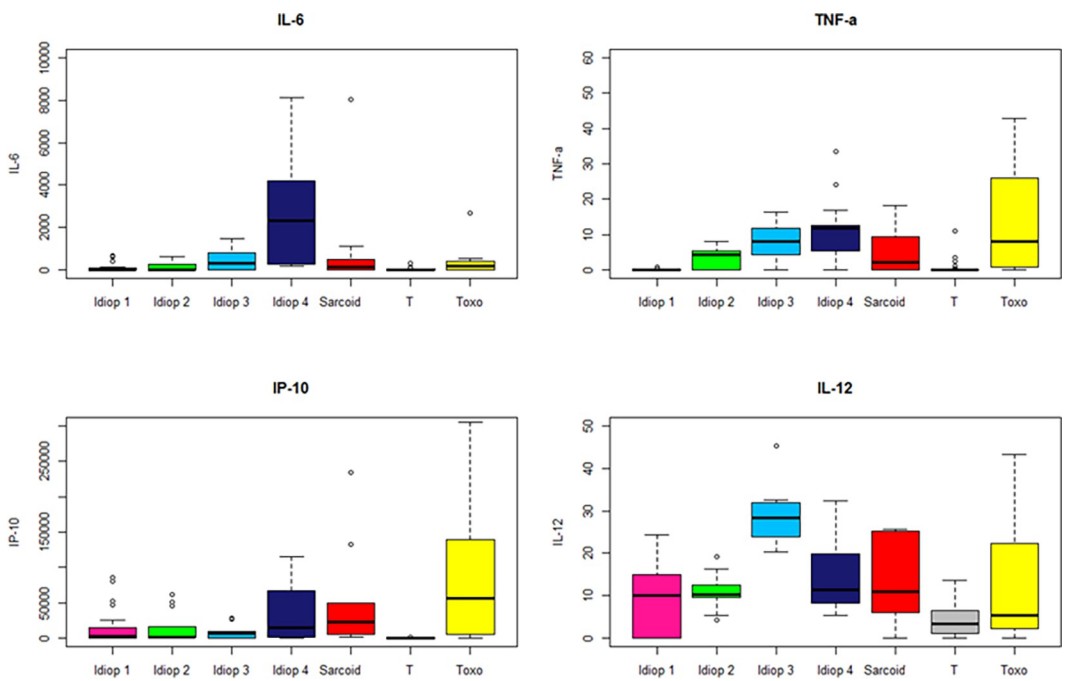

**Fig 10. Boxplots of 4 immune mediators significantly elevated in the aqueous humor of patients within 4 sub-groups of idiopathic uveitis (Idiop 1, 2, 3 and 4): IL-6, TNF-α, IL-12 and IP-10, as compared to immune mediators in the aqueous humor of patients with uveitis related to sarcoidosis, TU (Toxo) and noninflammatory controls (T).** Significant P values are noted in the upper part of each graph. A P value < 0.05 was considered significant.

showed that IL-10 levels were similar to the noninflammatory controls' [12]. IL-10 is an anti-inflammatory cytokine that supresses the expression of pro-inflammatory chemokines and cytokines TNF-α, IFN-γ and IL-1β [41], adhesion molecules, as well as antigen-presenting and costimulatory molecules in monocytes/macrophages, neutrophils, and T cells [42]. A previous study has shown IL-10 elevated in 3 among the 22 AH samples from idiopathic uveitis that might reflect the immunoregulator role of this cytokine [12]. This corresponds to a control of inflammatory process in correlation with elevated levels of IL-6 [17, 40, 43].

We found the median levels of IL-9 (a Th9 cell-specific cytokine) elevated in the AH of idiopathic uveitis. Th9 cells are a sub-population of CD4+ T cells and they produce IL-9 and IL-10 in the presence of IL-4 and TGF-β [44, 45]. Mouse models have indicated a role for Th9 cells in immunity and immune-mediated disease and recent studies have indicated that IL-9-producing cells contribute to a group of autoimmune and chronic inflammatory disorders including systemic lupus erythematosus (SLE), multiple sclerosis (MS), inflammatory bowel diseases (IBD), rheumatoid arthritis (RA), asthma and psoriasis [46–48]. To our knowledge we are the fist to report increased IL-9 levels in HA in idiopathic uveitis.

- **Other chemokines and cytokines**

A previous study has shown a positive correlation between the different cytokines, chemokines being elevated (IL-6, IL-8, MCP-1, and IFN-γ), in the AH of patients with idiopathic uveitis [12]. We confirmed an increase of median levels of IL-6, IL-8 and MCP-1.

We confirmed the findings of elevated median levels of IL-8 in 15 AH samplings from patients with idiopathic uveitis as previously demonstrated in intermediate uveitis of idiopathic origins [12] but also in Behcet disease and in active noninfectious uveitis [49]. IL-8 has a role in the neutrophils activation and in leukocytes migration into inflamed tissues.

**Table 4. Immune mediator expression (cytokines, chemokines and growth factors) elevated in patients with idiopathic uveitis and disease control groups (infectious and noninfectious uveitis) as compared with noninflammatory controls (*n* = number of patients).** The numbers in brackets represent the number of samples for which the mediator result was found to be high.

| Infectious uveitis | | | |
|---|---|---|---|
| **Toxoplasmosis (TU)** | **27 cytokines** | **serum (*n* = 16)** | **AH (*n* = 11)** |
| | | IL-2, GM-CSF (**1**) | G-CSF, IFN-γ, IP-10 (**9**)<br>eotaxin (**7**)<br>MCP-1 et TNF-α (**5**)<br>IL-8 (**4**)<br>IL-4 (**3**)<br>IL-1β, IL-6, IL-9, IL-10, IL-12, IL-13, MIP-1a, PDGF-BB, VEGF (**2**)<br>IL-1Ra, IL-2, IL-17, RANTES, VEGF (**1**) |
| | cytokines IL-21, IL-23 | serum (*n* = 14) | AH (*n* = 3) |
| | | none | IL-21 (**1**) |
| **noninfectious uveitis** | | | |
| **sarcoidosis** | 27 cytokines | serum (*n* = 12) | AH (*n* = 15) |
| | | IL-1β, IL-1Ra, IL-2, IL-6, éotaxine, G-CSF, GM-CSF, IP-10, IFN-γ, MIP-1b, TNF-α (**1**) | IP-10 (**14**)<br>MCP-1, eotaxine (**7**)<br>MIP-1a,<br>IL-8, G-CSF (**6**)<br>TNF-α (**4**)<br>IFN-γ (**3**)<br>IL-1β, IL-7, MIP-1b et PDGF-BB (**2, 4**)<br>IL-1Ra, IL-4, IL-10, IL-9, IL-13, IL-15, VEGF (**1**) |
| | cytokines IL-21, IL-23 | serum (*n* = 14) | AH (*n* = 3) |
| | | none | none |
| **Behcet** | 27 cytokines | serum (*n* = 7) | AH (*n* = 5) |
| | | IL-1β (**2**) | IP-10 (**5**)<br>IL-6, IL-8, G-CSF, MCP-1 (**4**)<br>IL-1β, IL-1Rα, IL-7, IL-9, IL-12, IL-13, eotaxin, IFN-γ, MIP-1α, PDGF-BB, TNF-α (**3**) |
| | cytokines IL-21, IL-23 | serum (*n* = 6) | AH (*n* = 1) |
| | | none | none |
| **idiopathic uveitis** | 27 cytokines | serum (*n* = 63) | AH (*n* = 69) |
| | | IL-7 (**7**)<br>PDGF-BB (**5**)<br>IL-5, IL-6 (**4**)<br>IL-1Rα(**3**)<br>IL-1β, IL-2, IL-4, IL-10, IL-12, GM-CSF, VEGF (**2**)<br>IL-15, G-CSF, IFN-γ, MIP-1α, MIP-1β, RANTES, TNF-α (**1**) | IL-12 (**36**)<br>IP-10 (**32**)<br>G-CSF (**29**)<br>IL-5 (**26**)<br>IL-10 (**22**)<br>VEGF (**21**), MCP-1 (**18**)<br>IL-6 (**15**), IL-8, TNF-α (**14**)<br>IL-13 (**12**), FGF-b (**10**), IL-17 (**10**)<br>IL-12, MIP-1α, GM-CSF, PDGF-BB (**8**)<br>IL-1Rα (**5**), MIP-1β, (**7**), RANTES (**6**)<br>IL-2 (**4**), IL-4, IFN-γ (**3**), IL-1β (**2**),<br>IL-7 (**1**) |
| | cytokines IL-21, IL-23 | serum (*n* = 68) | AH (*n* = 58) |
| | | IL-21 (**38**)<br>IL-23 (**6**) | IL-21, IL-23 (**16**) |

Moreover, IL-8 induces the differentiation of mononuclear cells into infiltrative granulocytes and the adhesion of leukocytes from the peripheral blood to the endothelial cells. In previous studies, the increase of IL-8 in the ocular form of Behcet disease has been shown as having the role to attract the polymorphonuclear neutrophils towards the lesions [50, 51].

The median levels of IL-6 were elevated in the AH from patients with idiopathic uveitis. This is in agreement with previous studies that showed an increase of IL-6 in the anatomical types of anterior uveitis [12, 52, 53]. Herein, we were able to demonstrate that increase of IL-6 also in intermediate, posterior and panuveitis. IL-6 is pleiotropic and proinflammatory produced by T cells, monocytes, macrophages and synovial fibroblastes. This cytokine is involved in the Th17 cells differentiation by regulating the balance between Th17 lymphocytes and Treg cells and is also involved in suppressing the differentiation [54]. IL-6, IL-8 and MCP-1 have been shown as regulated by the nuclear factor NF-kB pathway that plays a key role in the immune response [55]. Previous study have shown increased IL-6 and IL-8 in the intraocular samples of patients with TU, viral uveitis, Fuchs iridocyclitis, ocular Behcet disease and pediatrics uveitis [12, 56].

We found G-CSF median levels elevated in the AH of patients with idiopathic uveitis. Increased G-CSF have also been found in serum and synovial fluid of patients rheumatoid arthritis and correlated with disease severity [57]. Adding G-CSF increases the number of neutrophils in the serum and the endogenous G-CSF is important for the basal granulopoiesis. The infiltration of target tissues by the recruitment of neutrophils during inflammation is characteristic in both acute and chronic settings and the leukocytes population is mostly polynuclear neutrophils found in the inflamed joints in rheumatoid arthritis, for instance [58].

We found median levels of MCP-1 (CCL-2) elevated in the AH of patients with idiopathic uveitis in our study. MCP-1 is one of the key chemokines that regulate migration and infiltration of monocytes/macrophages into foci of active inflammation [59].

We found in the AH from patients with idiopathic uveitis, an elevated median level of IL-5. IL-5 is a cytokine produced by Th2 activated lymphocytes and mastocytar cells that selectively stimulate the differentiation, proliferation and fonctionnal activation of eosinophils. In Takase et al's study, IL-5 was detected in the AH samples from patients with viral acute retinal necrosis and in patients with anterior uveitis related to herpesvirus. IL-5 was not detected in noninfectious uveitis [17].

In our study IL-1β was elevated in 3 out of 69 samples from patients with idiopathic uveitis only (4%). IL-1β acts locally like an amplification signal in the pathological process associated with chronic inflammation as show previously in the vitreous from patients with idiopathic panuveitis [60].

As regards to the chemokines and cytokines found in the serum, some isolated patients with idiopathic uveitis had some other mediators elevated apart from IL-17, IP-10 and IL-21. Those mediators were the following ones: IL-1β, IL-1Rα, IL-2, IL-4, IL-6, IL-7, IL-10, IL-12, IL-15, IFN-γ, G-CSF, MIP-1α, MIP-1β, TNF-α, RANTES, PDGF-BB and VEGF, meaning that various sub-groups in idiopathic uveitis might exist (Table 4, Supplemental data). Yet, a major obstacle for using anti-VEGF intraocular treatments for inflammatory macular edema for therapeutic targeting is the notoriously lack of dosing VEGF in the ocular samples in clinical practice. This implies the hypothesis that VEGF may contribute to the development of uveitic macular edema. Several small retrospective and prospective studies have shown a moderate reduction of macular thickness and the need for an ongoing phase III randomized parallel design trial (MERIT study (NCT02623426). Interestingly, we found 21 samples of AH with VEGF elevated out of 69 samples of AH in idiopathic uveitis. More studies are needed to determine if the optimal efficacy of anti-VEGF treatments is found in the eyes where VEGF levels are increased. Our study shows that this rate is not increased in all AH samples.

The limitation of the study is that the degree of inflammation in the anterior segment was lower in idiopathic uveitis as compared as ocular sarcoidosis and ocular toxoplasmosis. Moreover, we have analyzed the aqueous humor which is more convenient to get in routine practice than vitreous humor. The use of aqueous humor allowed us to analyze what is to our knowledge the largest series published of 75 aqueous humors from eyes affected by idiopathic uveitis. It has been in a previous paper that the mediators are present in significantly higher concentrations the vitreous humor in the 2–4 cell group than in the 0–1 cell group (according to cells in anterior chamber and in vitreous), whereas those of IL-10 and IL-26 were significantly higher in the 0–1 cell group [61].

## Conclusion

Noninfectious uveitis represents a clinically heterogeneous set of ocular diseases that share immune characteristics with systemic auto-inflammatory conditions [62–65].

The main aim of the study was to explore the cytokines/ chemokines/ growth factors profile of noninfectious uveitis to better understand its enigmatic etiology and lay the groundwork for emerging anti-cytokines based therapeutics (biologics) and anti-VEGF treatments.

Unbiased computational mining of multiplex immunoassay data identified 4 clusters of mediators (cytokines/ chemokines and growth factors) characterized by the order of magnitude of concentrations of intraocular cytokines.

Idiopathic uveitis in humans has long been considered as a Th1-mediated disease, with interferon (IFN)-γ and IL-12 as signature cytokines. We confirm those findings in idiopathic uveitis of the intermediate and posterior anatomical types because we found IFN-γ increased in the AH samples tested. In several causes of uveitis however, it has been shown that both innate and adaptive immunity, leading to the activation of the IL-23/Th17 axis, may contribute to the initiation of tissue inflammation. The inflammatory mechanisms in the cases of idiopathic uveitis are possibly mediated by the Th17 pathway because we demonstrated IL-17 elevated in both AH and serum and IL-21 elevated in the serum.

In our present study, IL-9 has been also demonstrated to be probably involved in the pathogenesis of idiopathic uveitis because we demonstrated IL-9 levels being elevated in the AH as it has been shown in other chronic inflammatory disease but also of auto-immune pathogenesis.

We also present data on increased ocular concentrations of IFN-γ-inducing cytokine (IL-12) and IFN-γ-inducible CXC chemokine (IP-10), suggesting that IFN plays a central role in cellular immunity. In our study, this hypothesis was also suspected by the statistical analysis of hierarchical unsupervised classification, that described 4 subgroups (or clusters) of idiopathic uveitis characterized by the order of magnitude of concentrations of intraocular cytokines.

Other proinflammatory (IL-6, IL-8, IP-10, MCP-1, G-CSF, MIP-1α et MIP-1β et TNF-α) and antiinflammatory (IL-1Rα) immune mediators were increased in AH samples from idiopathic uveitis. Moreover, we also demonstrated in the serum samples of patients with idiopathic uveitis elevated levels of IL-17, IL-21 and IP-10 that might imply a systemic immune mechanism for some of them instead of a purely ocular limited disease as suggested by the clinical examination.

## Supporting information

**S1 Data.**
(XLSX)

**S2 Data.**
(XLSX)

## Acknowledgments

Dr Sabrina Falah, Dr Auclin, Dr Bottin, Dr Raphaelle Ores, Clémence Virevialle, Dr Jonathan Benesty, Dr Jad Akesbi, Dr Antoine Labbe, Dr Raphael Adam, Dr Tibault Rodallec, Dr Lyes Meziani, Ms Claire Poilane, Mr Didier Gleize, Miss Hélène Segara, Mr. Frédéric Diefhental

## Author Contributions

**Conceptualization:** Marie-Hélène Errera, Sylvain Fisson, Vincent Levy, Christine Chaumeil, José-Alain Sahel, Pablo Goldschmidt, Coralie Bloch-Queyrat.

**Data curation:** Marie-Hélène Errera, Thomas Manicom, Emmanuel Héron, Lilia Merabet, Alfred Kobal, Françoise Brignole-Baudouin, José-Alain Sahel, Coralie Bloch-Queyrat.

**Formal analysis:** Marie-Hélène Errera, Ana Pratas, Sylvain Fisson, Thomas Manicom, Marouane Boubaya, Neila Sedira, Emmanuel Héron, Lilia Merabet, Vincent Levy, Christine Chaumeil, Françoise Brignole-Baudouin, Pablo Goldschmidt, Bahram Bodaghi, Coralie Bloch-Queyrat.

**Funding acquisition:** Marie-Hélène Errera.

**Investigation:** Marie-Hélène Errera, Ana Pratas, Neila Sedira, Lilia Merabet, Alfred Kobal, Christine Chaumeil, Pablo Goldschmidt.

**Methodology:** Marie-Hélène Errera, Ana Pratas, Sylvain Fisson, Thomas Manicom, Marouane Boubaya, Alfred Kobal, Vincent Levy, Christine Chaumeil, Françoise Brignole-Baudouin, Pablo Goldschmidt, Bahram Bodaghi, Coralie Bloch-Queyrat.

**Project administration:** Marie-Hélène Errera, Neila Sedira, Coralie Bloch-Queyrat.

**Resources:** Marie-Hélène Errera, Vincent Levy, Christine Chaumeil, Françoise Brignole-Baudouin, José-Alain Sahel.

**Software:** Ana Pratas, Marouane Boubaya, Vincent Levy, Coralie Bloch-Queyrat.

**Supervision:** Marie-Hélène Errera, Sylvain Fisson, Lilia Merabet, Vincent Levy, Jean-Michel Warnet, Christine Chaumeil, Françoise Brignole-Baudouin, José-Alain Sahel, Pablo Goldschmidt, Bahram Bodaghi, Coralie Bloch-Queyrat.

**Validation:** Marie-Hélène Errera, Sylvain Fisson, Emmanuel Héron, Jean-Michel Warnet, Christine Chaumeil, Françoise Brignole-Baudouin, José-Alain Sahel, Pablo Goldschmidt, Bahram Bodaghi, Coralie Bloch-Queyrat.

**Visualization:** Marie-Hélène Errera, Jean-Michel Warnet.

**Writing – original draft:** Marie-Hélène Errera.

**Writing – review & editing:** Marie-Hélène Errera, Neila Sedira, Emmanuel Héron, Lilia Merabet, Alfred Kobal, Vincent Levy, Jean-Michel Warnet, Christine Chaumeil, Françoise Brignole-Baudouin, José-Alain Sahel, Pablo Goldschmidt, Bahram Bodaghi, Coralie Bloch-Queyrat.

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
