## [Decision Letter · Decision Letter 0]

6 Aug 2021

PONE-D-21-21937

IMMMUNE MEDIATORS IN IDIOPATHIC UVEITIS

PLOS ONE

Dear Dr. Errera,

Thank you for submitting your manuscript to PLOS ONE. After careful consideration, we feel that it has merit but does not fully meet PLOS ONE’s publication criteria as it currently stands. Therefore, we invite you to submit a revised version of the manuscript that addresses the points raised during the review process.

We look forward to receiving your revised manuscript.

Kind regards,

Ashok Kumar, Ph.D.

Academic Editor

PLOS ONE

Journal Requirements:

2. Please modify the title to ensure that it is meeting PLOS’ guidelines (https://journals.plos.org/plosone/s/submission-guidelines#loc-title). In particular, the title should be "specific, descriptive, concise, and comprehensible to readers outside the field" and in this case we feel it is not informative and specific about your study's scope and methodology.

Additional Editor Comments (if provided):

The manuscript has been reviewed by two experts in the field of uveitis. Although they found the study being important, they raised significant concerns. Please revise the manuscript and address concerns by point-by-point responses.

Reviewers' comments:

Reviewer's Responses to Questions

**Comments to the Author**

1. Is the manuscript technically sound, and do the data support the conclusions?

Reviewer #1: Partly

Reviewer #2: No

2. Has the statistical analysis been performed appropriately and rigorously? 

Reviewer #1: No

Reviewer #2: No

3. Have the authors made all data underlying the findings in their manuscript fully available?

Reviewer #1: Yes

Reviewer #2: No

4. Is the manuscript presented in an intelligible fashion and written in standard English?

Reviewer #1: Yes

Reviewer #2: No

5. Review Comments to the Author

Reviewer #1: The manuscript by Marie-Hélène Errera et al reported cytokines, chemokines and growth factors deregulated in idiopathic uveitis. Serum and aqueous humor (AH) samples from patients with idiopathic uveitis were compared with samples from patients with ocular toxoplasmosis, patients with Behçet disease related uveitis and patients with sarcoidosis related uveitis. A control group (patients with age-related cataract) was also included.

MAJOR COMMENTS

1) Authors stated: “The aim of this study was to investigate which cytokine, chemokines and growth factors are involved in the immunopathogenesis of idiopathic uveitis and whether specific cytokines

profiles are associated with clinical manifestations”. The reviewer thinks that the study design did not allow to identify which factors are involved in the immunopathogenesis of idiopatic uveitis. The study design allowed to identify immune mediators deregulated in idiopatic uveitis. Functional analyses are needed to identify their role. Please comment and discuss this concept in the manuscript. If possible provide some functional data.

2) Features used for the diagnosis of uveitis at the moment of sample withdrawal should be better detailed.

3) Materials and methods explaining the quantification of the cytokines are not clear. They would not be reproducible in the current version. Which kind of assay (from Bio-Rad or Invitrogen) was used? Which were the limits of detection? Were the AH samples centrifuged? At line 174 “culture supernatant” is mentioned. Is this correct?

4) Statistical analysis. When a cytokine was not detected in all the samples, Fisher test should also be applied. Moreover, the reviewer suggests to use Kruskal-Wallis test to compare cytokine levels among different groups of patients.

5) The degree of inflammation in anterior segment was low (Table 3). Could this explain same data? Which was the degree of inflammation in the other patients (ocular toxoplasmosis, Behçet disease, sarcoidosis)?

6) The reviewer thinks that Table 4 is scarcely informative. What did the numbers in brackets represent?

MINOR COMMENTS

1) Table 1 should be modified to be clearer. Instead of “number analyzed” the reviewer thinks that the kind of assays (number of samples analyzed) are shown. If this is correct please modify accordingly. A small number of samples was tested for IL-21 & IL-23

Reviewer #2: In this work, the authors detected immune mediators in AH and serum of patients with idiopathic uveitis. Here are my concerns that need fixing.

1. Numerous writing errors should be corrected.

2. It seems that the authors are not familiar with paper organizing and writing, the structure of the whole paper need to improve.

3. More details of the patients should be provide, most importantly, such as the inflammatory grade of the patients while the AH and serum samples were taken.

4. Since idiopathic uveitis is only an exclusive diagnosis and it is an umbrella of uveitis with different immune pathology, the scientific significance of this research work is low.

6. PLOS authors have the option to publish the peer review history of their article (what does this mean?). If published, this will include your full peer review and any attached files.

Reviewer #1: No

Reviewer #2: No

---

## [Author Response · Author response to Decision Letter 0]

17 Oct 2021

• Journal requirement:

We have modified the styles as requested

2. Please modify the title to ensure that it is meeting PLOS’ guidelines (https://journals.plos.org/plosone/s/submission-guidelines#loc-title). In particular, the title should be "specific, descriptive, concise, and comprehensible to readers outside the field" and in this case we feel it is not informative and specific about your study's scope and methodology.

We have modified the title for: ”Cytokines, chemokines and growth factors profile in human aqueous humor in idiopathic uveitis”

The corresponding author has added an ORCID ID number.

• Reviewer #1:

Authors stated: “The aim of this study was to investigate which cytokine, chemokines and growth factors are involved in the immunopathogenesis of idiopathic uveitis and whether specific cytokines

profiles are associated with clinical manifestations”. The reviewer thinks that the study design did not allow to identify which factors are involved in the immunopathogenesis of idiopathic uveitis. The study design allowed to identify immune mediators deregulated in idiopathic uveitis. Functional analyses are needed to identify their role. Please comment and discuss this concept in the manuscript. If possible provide some functional data.”

We have modified the aim of the study as follow page 4 line 97: “ The aim of this study was to identify which cytokine, chemokines and growth factors are deregulated in idiopathic uveitis and whether specific cytokines profiles are associated with clinical manifestations.”

We have not performed any functional analysis of the CD4(+) T-cell for instance in our study.

2) Features used for the diagnosis of uveitis at the moment of sample withdrawal should be better detailed.

We have modified accordingly as follows :

“The diagnosis of uveitis at the moment of sample withdrawal was based on an ophthalmic examination consisting of visual acuity recordings (Snellen scale), slit-lamp examination, intraocular pressure measurement, and dilated fundus examination with indirect ophthalmoscopy. Ancillary testing such as fluorescein angiography and optical coherence tomography were performed when necessary. The classification of uveitis used was the anatomical classification (the International Uveitis Study Group (IUSG) [8]. The criteria for idiopathic (or unknow etiologies) were investigations oriented by the anatomic characteristics of uveitis: negative serologic screening for syphilis, normal serum angiotensin-converting enzyme, and interferon-gamma release, normal chest computed tomography. Our group has published a standardized strategy that we use in routine for the etiologic diagnosis of uveitis with first (CBC, ESR, CRP, quantiferon, syphilis serology, chest radiograph), second (ACE, antinuclear antibodies, complement, HLA B27 etc…) and third steps investigations based on the clinical type of uveitis and clinical and medical history findings. A cerebral magnetic resonance imaging and anterior chamber tap with interleukin-10 analysis and cytology, Herpes viridea (HSV, VZV, CPV) PCR and/or Goldmann coefficient are part of the second/ third steps investigations for chronic intermediate, posterior and panuveitis or when severe and/or corticoresistant uveitis [9]. We excluded patients based any past history of systemic inflammation, auto-immune disease, concomitant anti-inflammatory treatment, immunosuppressed state or systemic antibiotics or immunomodulatory therapy within 4 weeks before inclusion.”

3) Materials and methods explaining the quantification of the cytokines are not clear. They would not be reproducible in the current version. Which kind of assay (from Bio-Rad or Invitrogen) was used? Which were the limits of detection? Were the AH samples centrifuged? At line 174 “culture supernatant” is mentioned. Is this correct?

We used the Luminex methodology® specially calibrated and validated for microdosing cytokines and chemokines in intraocular fluids (flow microcytometry combining the use of microspheres fluorescents recognizing peptides and a double reading after excitation by two lasers)

The dynamic range for Luminex assays is ∼1-10,000 pg/mL (Elshal MF, McCoy JP. Multiplex bead array assays: performance evaluation and comparison of sensitivity to ELISA. Methods 2006;38:317–23; Dupont NC, Wang K, Wadhwa PD, Culhane JF, Nelson EL. Validation and comparison of luminex multiplex cytokine analysis kits with ELISA: determinations of a panel of nine cytokines in clinical sample culture supernatants. J Reprod Immunol 2005;66:175–91; Ray CA, Bowsher RR, Smith WC, et al. Development, validation, and implementation of a multiplex immunoassay for the simultaneous determination of five cytokines in human serum. J Pharm Biomed Anal 2005;36:1037–44.)

For serological examinations of cytokines, the blood was taken from a tube without anticoagulant and centrifuged for 15 minutes at 4 °C and the serum frozen at -80 ° C and thawed immediately before biological analysis.

Sharma RK, Rogojina AT, Chalam KV. Multiplex immunoassay analysis of biomarkers in clinically accessible quantities of human aqueous humor. Mol Vis. 2009;15:60-9. Epub 2009 Jan 14. PMID: 19145248; PMCID: PMC2622713.

Culture supernatant was a typographic error, we removed it.

4) Statistical analysis. When a cytokine was not detected in all the samples, Fisher test should also be applied. Moreover, the reviewer suggests to use Kruskal-Wallis test to compare cytokine levels among different groups of patients.

We have incorporated your suggestions in our statistical analysis and therefore we have used Fisher and Kruskal-Wallis tests to compare cytokine levels among different groups of patients. P values were modified in the Result section of the manuscript and in Table 2, accordingly. Thank for your suggestions, they were very useful.

5) The degree of inflammation in anterior segment was low (Table 3). Could this explain same data? Which was the degree of inflammation in the other patients (ocular toxoplasmosis, Behçet disease, sarcoidosis)?

Thank you for your comment, this is useful. 

We have added a comment in discussion as follow: “The limitation of the study is that the degree of inflammation in the anterior segment was lower in idiopathic uveitis as compared as ocular sarcoidosis and ocular toxoplasmosis. Moreover, we have analyzed the aqueous humor which is more convenient to get in routine practice than vitreous humor. The use of aqueous humor allowed us to analyze what is to our knowledge the largest series published of 75 aqueous humors from eyes affected by idiopathic uveitis. It has been in a previous paper that the mediators are present in significantly higher concentrations the vitreous humor in the 2–4 cell group than in the 0–1 cell group (according to cells in anterior chamber and in vitreous), whereas those of IL-10 and IL-26 were significantly higher in the 0–1 cell group [Fukunaga H, Kaburaki T, Shirahama S, Tanaka R, Murata H, Sato T, Takeuchi M, Tozawa H, Urade Y, Katsura M, Kobayashi M, Wada Y, Soga H, Kawashima H, Kohro T, Aihara M. Analysis of inflammatory mediators in the vitreous humor of eyes with pan-uveitis according to aetiological classification. Sci Rep. 2020 Feb 17;10(1):2783]”

We have incorporated your suggestions and therefore we have added the inflammatory grade in the anterior segment and vitreous in eyes with intermediate uveitis, sarcoidosis, Behcet disease and toxoplasmosis in Table 1.

6) The reviewer thinks that Table 4 is scarcely informative. What did the numbers in brackets represent?

We agree with the reviewer and therefore we have moved Table 4 in supplemental data. The numbers in brackets represent the number of samples for which the mediator result was found to be high.

MINOR COMMENTS

1) Table 1 should be modified to be clearer. Instead of “number analyzed” the reviewer thinks that the kind of assays (number of samples analyzed) are shown. If this is correct please modify accordingly. A small number of samples was tested for IL-21 & IL-23

We have modified Table 1 accordingly. IL-21 and IL-23 were in a separate Luminex® panel and that testing was performed when enough aqueous humor volume was left after the first cytokines panel (IL-1β, IL-2, IL-6, IL-12p70, IL-17, interferon-γ….) was performed. 

• Reviewer #2: 

1. Numerous writing errors should be corrected.

We have corrected the writing errors. We are thankful for the Reviewer comment.

2. It seems that the authors are not familiar with paper organizing and writing, the structure of the whole paper need to improve.

We have incorporated your suggestions and therefore we have modified the organizing and writing of the whole paper. Thank for your suggestions, they were very useful.

3. More details of the patients should be provided, most importantly, such as the inflammatory grade of the patients while the AH and serum samples were taken.

We have incorporated your suggestions and therefore we have added the inflammatory grade in the anterior segment and vitreous in eyes with intermediate uveitis, sarcoidosis, Behcet disease and toxoplasmosis in Table 1. Thank for your suggestions, they were very useful.

4. Since idiopathic uveitis is only an exclusive diagnosis and it is an umbrella of uveitis with different immune pathology, the scientific significance of this research work is low.

We agree with the Reviewer statement. However, we would like to emphasize the important points and the new lines of research that this manuscript has brought in particular on VEGF and the data on increased ocular concentrations of IFN-γ-inducing cytokine (IL-12) and IFN-γ-inducible CXC chemokine (IP-10), suggesting that IFN plays a central role in cellular immunity.

Below, below the Reviewer will find the modifications made to the conclusion:

Noninfectious uveitis represents a clinically heterogeneous set of ocular diseases that share immune characteristics with systemic auto-inflammatory conditions (Lee RW, Nicholson LB, Sen HN, et al. Autoimmune and autoinflammatory mechanisms in uveitis. Semin Immunopathol. 2014;36:581–594; Caspi RR. A look at autoimmunity and inflammation in the eye. J Clin Invest. 2010;120:3073–3083; Wakefield D, Yates W, Amjadi S, McCluskey P. HLA-B27; anterior uveitis: immunology and immunopathology. Ocul Immunol Inflamm. 2016;24:450–459; Mesquida M, Molins B, Lloren¸c V, de la Maza MS, Ad´an A; Targeting interleukin-6 in autoimmune uveitis. Autoimmun. Rev. 2017;16:1079–1089). Unbiased computational mining of multiplex immunoassay data identified 4 clusters of mediators (cytokines/ chemokines and growth factors) characterized by the order of magnitude of concentrations of intraocular cytokines.

The main aim of the study was to explore the cytokines/ chemokines/ growth factors profile of noninfectious uveitis to better understand its enigmatic etiology and lay the groundwork for emerging anti-cytokines based therapeutics (biologics) and anti-VEGF treatments. 

Yet, a major obstacle for using anti-VEGF intraocular treatments for inflammatory macular edema for therapeutic targeting is the notoriously lack of dosing VEGF in the ocular samples in clinical practice. This implies the hypothesis that VEGF may contribute to the development of uveitic macular edema. Several small retrospective and prospective studies have shown a moderate reduction of macular thickness and the need for an ongoing phase III randomized parallel design trial (MERIT study (NCT02623426). Interestingly, we found 21 samples of AH with VEGF elevated out of 69 samples of AH in idiopathic uveitis. More studies are needed to determine if the optimal efficacy of anti-VEGF treatments is found in the eyes where VEGF levels are increased. Our study shows that this rate is not increased in all AH samples.

We also present data on increased ocular concentrations of IFN-γ-inducing cytokine (IL-12) and IFN-γ-inducible CXC chemokine (IP-10), suggesting that IFN plays a central role in cellular immunity. In our study, this hypothesis was also suspected by the statistical analysis of hierarchical unsupervised classification, that described 4 subgroups (or clusters) of idiopathic uveitis characterized by the order of magnitude of concentrations of intraocular cytokines.

---

## [Decision Letter · Decision Letter 1]

11 Nov 2021

Cytokines, chemokines and growth factors profile in human aqueous humor in idiopathic uveitis

PONE-D-21-21937R1

Dear Dr. Errera,

We’re pleased to inform you that your manuscript has been judged scientifically suitable for publication and will be formally accepted for publication once it meets all outstanding technical requirements.

Kind regards,

Ashok Kumar, Ph.D.

Academic Editor

PLOS ONE

Additional Editor Comments (optional):

Authors have addressed prior concerns raised by both reviewers. The manuscript is much improved.

Reviewers' comments:

Reviewer's Responses to Questions

**Comments to the Author**

1. If the authors have adequately addressed your comments raised in a previous round of review and you feel that this manuscript is now acceptable for publication, you may indicate that here to bypass the “Comments to the Author” section, enter your conflict of interest statement in the “Confidential to Editor” section, and submit your "Accept" recommendation.

Reviewer #1: All comments have been addressed

Reviewer #2: (No Response)

2. Is the manuscript technically sound, and do the data support the conclusions?

Reviewer #1: Yes

Reviewer #2: Partly

3. Has the statistical analysis been performed appropriately and rigorously? 

Reviewer #1: Yes

Reviewer #2: Yes

4. Have the authors made all data underlying the findings in their manuscript fully available?

Reviewer #1: Yes

Reviewer #2: Yes

5. Is the manuscript presented in an intelligible fashion and written in standard English?

Reviewer #1: Yes

Reviewer #2: No

6. Review Comments to the Author

Reviewer #1: Authors answered in a good way the comments of the reviewer. The manuscript has improved and is clearer.

Reviewer #2: (No Response)

7. PLOS authors have the option to publish the peer review history of their article (what does this mean?). If published, this will include your full peer review and any attached files.

Reviewer #1: **Yes: **Stefania Croci

Reviewer #2: No

---

## [Editor Report · Acceptance letter]

5 Jan 2022

PONE-D-21-21937R1 

Cytokines, chemokines and growth factors profile in human aqueous humor in idiopathic uveitis 

Dear Dr. Errera:

I'm pleased to inform you that your manuscript has been deemed suitable for publication in PLOS ONE. Congratulations! Your manuscript is now with our production department. 

Kind regards, 

on behalf of

Dr. Ashok Kumar 

Academic Editor

PLOS ONE